# Comparison of Robotic and Conventional Unicompartmental Knee Arthroplasty Outcomes in Patients with Osteoarthritis: A Retrospective Cohort Study

**DOI:** 10.3390/jcm11010220

**Published:** 2021-12-31

**Authors:** Christopher Wu, Nobuei Fukui, Yen-Kuang Lin, Ching-Yu Lee, Shih-Hsiang Chou, Tsung-Jen Huang, Jen-Yuh Chen, Meng-Huang Wu

**Affiliations:** 1College of Medicine, Taipei Medical University, Taipei 110, Taiwan; cjwuchris@gmail.com; 2Department of Medicine, Taipei Medical University Hospital, Taipei 110, Taiwan; m.fukui21@gmail.com; 3Research Center of Statistics, College of Management, Taipei Medical University, Taipei 110, Taiwan; robbinlin@tmu.edu.tw; 4Department of Orthopedics, Taipei Medical University Hospital, Taipei 110, Taiwan; ejaca22@gmail.com (C.-Y.L.); tjdhuang@tmu.edu.tw (T.-J.H.); 5Department of Orthopedics, School of Medicine, College of Medicine, Taipei Medical University, Taipei 110, Taiwan; 6Department of Orthopedics, Kaoshiung Medical University Hospital, Kaoshiung 807, Taiwan; stanelychou@gmail.com; 7Regenerative Medicine and Cell Therapy Research Center, Kaoshiung Medical University, Kaoshiung 807, Taiwan; 8Orthopedic Research Center, Kaoshiung Medical University, Kaoshiung 807, Taiwan; 9Department of Orthopedics, Postal Hospital, Taipei 100, Taiwan

**Keywords:** unicompartmental knee arthroplasty, robotic arm, osteoarthritis, outcome

## Abstract

Robotic-arm-assisted unicompartmental knee arthroplasty (RUKA) was developed to increase the accuracy of bone alignment and implant positioning. This retrospective study explored whether RUKA has more favorable overall outcomes than conventional unicompartmental knee arthroplasty (CUKA). A total of 158 patients with medial compartment osteoarthritis were recruited, of which 85 had undergone RUKA with the Mako system and 73 had undergone CUKA. The accuracy of component positioning and bone anatomical alignment was compared using preoperative and postoperative radiograph. Clinical outcomes were evaluated using questionnaires, which the patients completed preoperatively and then postoperatively at six months, one year, and two years. In total, 52 patients from the RUKA group and 61 from the CUKA group were eligible for analysis. The preoperative health scores and Kellgren–Lawrence scores were higher in the RUKA group. RUKA exhibited higher implant positioning accuracy, thus providing a superior femoral implant angle, properly aligned implant placement, and a low rate of overhang. RUKA also achieved higher accuracy in bone anatomical alignment (tibial axis angle and anatomical axis angle) than CUKA, but surgical time was longer, and blood loss was greater. No significant differences were observed in the clinical outcomes of the two procedures.

## 1. Introduction

Unicompartmental knee arthroplasty (UKA) is a reliable treatment for patients with osteoarthritis limited to either the medial or lateral compartment of the knee [1]. UKA is a procedure in which the damaged compartment is replaced with an implant prosthesis. This procedure preserves the nonaffected bone and ligament and is, therefore, a more widely used alternative to total knee arthroplasty (TKA) [2,3]. UKA has several advantages over TKA, such as a shorter operating time, less intraoperative blood loss, a lower complication rate, faster recovery, superior functional outcomes, and higher patient satisfaction [4]. However, conventional UKA (CUKA) is associated with a higher revision rate and lower implant survival than TKA [5]. Various factors, such as implant positioning, bone alignment, and soft-tissue balance, can affect surgical outcomes [6].

New robotic-arm-assisted technologies have been developed to increase accuracy and improve component positioning and bone anatomical alignment [7]. Two systems have been approved by the US Food and Drug Administration for robotic-arm-assisted UKA (RUKA) [8]. One is the Mako guided robot, which was introduced in 2005 and is widely used in UKA operations [9]. The Mako system uses computed tomography (CT) for preoperative mapping and provides a three-dimensional (3D) model of the patient’s knee. This system also provides intraoperative feedback on positioning, facilitating accurate implantation of the femoral and tibial components [4,9].

To the best of our knowledge, few studies have compared conventional and robotic-assisted surgical outcomes in patients with unicompartmental knee osteoarthritis. We hypothesized that RUKA would provide increased accuracy in component positioning and bone angle alignment, improved functional outcomes, less pain, and greater patient satisfaction than CUKA. Thus, in this retrospective study, we assessed outcomes in our patient groups.

## 2. Materials and Methods

A total of 158 patients with medial unicompartmental osteoarthritis who had undergone UKA were recruited for this study. Of them, 73 patients had undergone CUKA between March 2001 and June 2016, and 85 had undergone RUKA between July 2017 and December 2018. The mean patient age was 69.36 years (standard deviation [SD], 9.14) in the conventional group and 68.52 years (SD, 9.78) in the robotic arm group. All operations were performed by an experienced orthopedic surgeon at Postal Hospital (Taipei, Taiwan). Figure 1 presents a flowchart of the study.

Patients were excluded if they had osteoarthritis affecting lateral or multiple compartments, inflammatory arthritis related to underlying diseases, a history of other conditions in the lower extremities (e.g., tumors, fractures, infection, surgery, and avascular necrosis), bony deformities, or contraindications to the implantation of a device. In addition, of the patients initially recruited for the study, 12 who had undergone CUKA and 33 who had undergone RUKA were excluded because they required bilateral unicompartmental arthroplasty, lateral compartmental arthroplasty, or TKA. Consequently, 61 patients from the CUKA group and 52 from the RUKA group were eligible for the preoperative and postoperative analyses. Clinical outcomes were assessed through questionnaires, which were also used to record participant data, including chart number, phone number, age, sex, details of the knee affected, and surgery date. The questionnaires were the Numeric Rating Scale (NRS), the Western Ontario and McMaster Universities Arthritis Index (WOMAC), and the five-level EQ-5D (EQ5D5L). They were administered to the patients preoperatively and then postoperatively at the 6-month, 1-year, and 2-year follow-up to assess pain, function, complications, and satisfaction. However, 14 patients from the RUKA group and 23 from the CUKA group did not complete the questionnaires and were thus excluded. Consequently, 38 patients who had undergone RUKA and 38 who had undergone CUKA were eligible for the clinical outcome analysis.

For the Mako system RUKA, a preoperative CT image is used to create a 3D computer model of the patient’s knee, which helps the surgeon develop an appropriate preoperative plan according to the size and position of the components, bone anatomical alignment, and volume of bone requiring resection. The bone volume data are then used to create a 3D model, which provides visual feedback from track marker arrays fixed to the femur and tibia; any cutting outside the marked incision area is resisted by the robotic arm through tactile feedback. A Stryker implant, Restoris MCK partial knee implant system (Stryker, Kalamazoo, MI, USA), was used in the RUKA procedure [5]. The conventional surgical procedure was performed using standard manual instrumentation and a Zimmer implant, Zimmer Unicompartmental High Flex Knee System (ZUK; Zimmer, Winterthur, Switzerland).

All the patients underwent preoperative and postoperative radiograph to assess bone alignment and component positioning. The bone alignment parameters were the anatomical axis angle, femoral condyle angle, femoral implant angle, tibial axis angle, and tibial slope, which were used in this study’s analysis (Figure 2A,B). These anatomical measurements were calculated using a two-dimensional knee radiograph measurement. The anatomical axis was defined as the angle between the mechanical and anatomical femoral axis on the anteroposterior (AP) knee radiograph. The anatomical femoral axis is a line drawn proximally to distally in the intramedullary canal of the femur, whereas the mechanical femoral axis is a line drawn from the center of the femoral head to the center of the knee joint [5,10,11]. The femoral condyle angle was defined as the lateral angle between the anatomical femoral axis and a line tangential to both the distal femoral condyles on the AP knee radiograph. The femoral implant angle was defined as the angle between the second hook of the posterior femoral component and the surface line of the femoral bone on the lateral knee radiograph. The tibial axis angle was defined as the medial angle between the tibial shaft axis and a line tangential to both the tibial condyles on the AP knee radiograph. The tibial slope was defined as the angle between the tibial implant and the surface line of the tibia bone on the lateral knee radiograph, which was then subtracted from 90°.

The sizes of the implant, the femur, and the tibial implant were measured. The implant positioning was assessed in the knee radiograph, and the results were categorized as (1) aligned or not aligned and (2) overhanging or normal. A slight overhang or underhang was defined as tibial component coverage 1–2 mm above or below, respectively, the recommended tolerance. An overhang or underhang was defined as tibial component coverage 3 mm or more above or below, respectively, the recommended tolerance. The preservation of the tibial eminence, the ridge of bone in the tibia where the anterior cruciate ligament attaches, was assessed by evaluating the position of the tibial implant in relation to the tibial eminence. The tibial eminence was cut only if it was surpassed by a component. Radiographs of these assessments are provided in online Appendix A.

Kellgren-Lawrence classification was employed preoperatively to assess the severity of a patient’s osteoarthritis by using AP knee radiographs. Grades between 0 and 4 were assigned, with grades 0 and 4 indicating no and severe osteoarthritis, respectively [12]. The patient’s American Society of Anesthesiologists (ASA) physical status classification, surgical time, blood loss, complications, and preoperative and postoperative hemoglobin levels were evaluated.

The data were recorded by the author, and the analysis was performed by a statistician at our institution. The statistical analysis was conducted using R statistical software. A chi-square test and *t* test were used for the data analysis. A *p* value of <0.05 was considered statistically significant.

## 3. Results

### Outcomes

Table 1 summarizes the outcomes of the postoperative bone and implant alignment parameter analysis. RUKA had a significantly higher accuracy of implant alignment than CUKA (*p* < 0.001). Three parameters exhibited significant differences: the anatomical axis angle (2.81° vs. 5.87°, *p* < 0.001), the femoral implant angle (37.59° vs. 30.04°, *p* < 0.001), and the tibial axis angle (85.07° vs. 87.57°, *p* < 0.001). The implants were more aligned in the RUKA group (57.7% vs. 18%, *p* < 0.001), with the component positioning recorded as overhanging or slightly overhanging (3.8% + 3.8% vs. 44.3% + 8.2%, *p* < 0.001), normal (28.8% vs. 34.4%, *p* < 0.001), and underhanging or slightly underhanging (15.4% + 48.1% vs. 4.9% + 4.9%, *p* < 0.001). Figure 3A–E presents the distribution of the alignment parameter measurements.

The analysis demonstrated significant differences between RUKA and CUKA in terms of preoperative health score, Kellgren–Lawrence score, surgery time, and blood loss (Table 2 and Table 3). The preoperative health score was significantly higher in the RUKA group (71.6 vs. 65.6, *p* = 0.006), as was the Kellgren–Lawrence score (2.77 vs. 2.25, *p* < 0.001), and the surgical time of the RUKA group was significantly longer than that of the CUKA group (60.87 vs. 43.85 min, *p* < 0.001). Blood loss was greater in the RUKA group than in the CUKA group (257.71 vs. 201.2 mL).

The comparison revealed no significant differences in the other bone parameters or preoperative parameters between the two groups (postoperative femoral condyle angle: 79.15° vs. 79.28°, *p* = 0.933; postoperative tibial slope: 10.38° vs. 10.00°, *p* = 0.535) (Table 1). The proportion of tibial eminence saved was not significantly different (26.9% vs. 36.1%, *p* = 0.403). Table 3 lists the preoperative and postoperative questionnaire results regarding clinical outcomes. Although the scores for postoperative clinical outcomes in both the RUKA and CUKA groups were lower than the preoperative scores, no significant differences were noted between the two groups (Table 3).

## 4. Discussion

We compared the implant positioning, bone alignment, and clinical outcomes of the RUKA and CUKA procedures. We used five radiological parameters to measure the accuracy of implant and bone alignment, and for three of the five parameters (the anatomical axis, femoral implant, and tibial axis angles), significant differences were observed. This demonstrates the accuracy of bone and implant positioning in RUKA with the Mako system. Although the results are not identical to those of other studies, they are similar in that RUKA increased the accuracy of implant positioning [5,13,14,15,16]. A study by Bell et al. [5] demonstrated that the implantation of components within 2° of the target position, when using the Mako system, was achieved in a relatively large proportion of patients. The positions of the femoral sagittal and coronal component and tibial sagittal and axial component were significantly more accurate when the robotic procedure was used than when the conventional procedure was used [5]. Mofidi et al. [17] assessed the accuracy of component positioning by comparing preoperative and postoperative radiographs. The results revealed that more than 70% of the tibial and femoral components were inserted within 3° of the target position [17]. In our study, accuracy was also evaluated in terms of whether components were overhanging, normal, or underhanging on the basis of the recommended tolerances, as described in the Materials and Methods section. Normal positioning or an underhang were considered accurate alignment, and the total proportion of components positioned within recommended tolerances was greater in RUKA than in CUKA (92.3% vs. 47.4%, *p* < 0.001). Underhang components do not cause soft tissue problems; therefore, this can be interpreted as a more favorable outcome. By contrast, overhang and slight overhang are considered inaccurate component positioning. We observed more inaccurately positioned components in CUKA than in RUKA (44.3% + 8.2% vs. 3.8% + 3.8%, *p* < 0.001). In CUKA, more overhang could result in an increased risk of knee pain due to soft tissue impingement on the edges.

Our results demonstrated that surgical time was significantly longer in RUKA than in CUKA. This factor was also assessed in a meta-analysis performed by Zhang et al. [2]. The longer surgical time of RUKA’s may be attributed to its novelty, meaning surgeons must practice to familiarize themselves with this new technique. The surgical time may decrease as surgeons gain familiarity [7]. In our study, blood loss was greater in RUKA than in CUKA, which is consistent with the results of Khan et al. [18]. The blood loss in RUKA could also be associated with surgeons’ lack of familiarity with the technique. In addition, the operating surgeon in our study used a tourniquetless technique for both groups; this explains the longer surgical time in RUKA, which could lead to increased blood loss. The strength of our study is that the robotic and conventional UKA procedures were performed by the same orthopedic surgeon; therefore, a direct comparison of surgical performance between the two groups was possible.

The results of the questionnaires (NRS, WOMAC, and EQ5D5L) demonstrated no significant difference in clinical outcomes between the RUKA and CUKA groups. This is consistent with the findings of Gilmour et al. [19], who revealed no differences in clinical outcomes at a 2-year follow-up. Blyth et al. [20] observed improvements in early pain and early function scores of some patients (median pain scores were 55.4% lower in the robotic group than in the conventional group, *p* = 0.004), but no significant difference was observed one year postoperatively. One study indicated lower postoperative pain and higher functional scores in RUKA compared with CUKA [7]. However, in a retrospective study of 62 patients undergoing UKA (with either Mako or conventional procedures) by Hansen et al., one CUKA procedure was changed to a TKA procedure because of infection, and knee pain was considerably more prevalent in the robotic group at the 2-year follow-up [21]. Patient satisfaction did not differ between the two groups in our study, which is consistent with the findings of Batailler et al. [13].

In this study, one patient underwent arthroscopy because of persistent pain after RUKA. However, no complications such as revision, infection, or breakage of implants were observed. This is consistent with the findings of a retrospective study by Lonner et al. [22], in which no soft tissue damage, bone injury, or other complications were observed postoperatively in 1,064 patients undergoing UKA (with either Navio PFS or the Mako system). A meta-analysis by Zhang et al. [2] demonstrated that patients experienced significantly fewer complications after RUKA than after CUKA, but no difference in revision rate was observed. However, Mergenthaler et al. [23] reported a higher revision rate in the conventional group and no significant difference in complication rate. Although increased component accuracy was achieved in our study through the robotic procedure, the correlation between component outcomes and bone malalignment remains uncertain. Mergenthaler et al. [23] revealed that limb alignment and UKA survival were closely related, and a study by Battenberg et al. [24] indicated that implant survival at a 2.3-year follow-up was higher for those receiving robotic-assisted surgery than for those receiving CUKA.

In our study, RUKA resulted in improved preoperative health and increased Kellgren–Lawrence scores, indicating that osteoarthritis was more severe in the robotic group than in the conventional group. The severity of osteoarthritis and its effect on postoperative prognosis remains unclear, and additional studies should be conducted to identify this effect. No significant differences in preoperative bone alignment parameters or duration of pain were observed in this study.

This study has several limitations. First, the five parameters might not be identical to those used in other studies. The femoral implant angle was created specifically for this study. Although the majority of principles used to calculate the parameters were similar, a direct comparison of the parameter results with those of other studies is not possible. To address this limitation, as a work in progress, we proposed using a CT scan as a method of conducting measurements, as in the study by Spinarelli et al. [25]. Second, the number of patients assessed in terms of functional outcomes was small because only patients who completed the questionnaires were included in the analysis. This may have affected the accuracy of the items in the clinical outcomes assessment, namely, pain scores, complications, functional outcomes, and implant survival. Third, the implants used for the robotic and conventional procedures differed; thus, a direct comparison of implant size between the two procedures was not possible. Fourth, postoperative hemoglobin was not recorded in patients who underwent RUKA. Although blood loss was recorded, a lack of postoperative hemoglobin data complicated the identification of the severity of the patients’ anemia, if present. Fifth, the cost for both groups was not analyzed, which might have affected the patients’ satisfaction and reported outcomes. Therefore, subsequent studies should investigate cost and effectiveness. Finally, this was a retrospective and nonrandomized study; thus, recall bias may have existed.

Nevertheless, our study had several strengths. The number of patients analyzed for the parameter measurements was sufficient (113). In addition, the 2-year follow-up was sufficient to assess postoperative outcomes. Finally, the operations were performed by the same orthopedic surgeon in the same institution; thus, variance in surgical performance was reduced.

## 5. Conclusions

This study demonstrated that RUKA resulted in higher component positioning accuracy than CUKA. However, a longer surgical time and increase in blood loss were observed in the RUKA group. No significant differences in clinical outcomes were observed between the two groups. Therefore, a follow-up study may be required to determine whether the increased accuracy of component positioning in RUKA improves clinical outcomes.

## Figures and Tables

**Figure 1 jcm-11-00220-f001:**
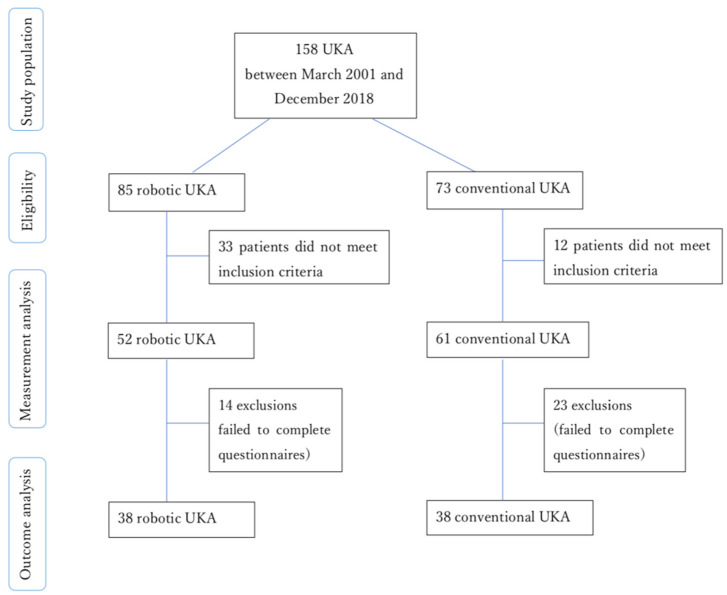
Flowchart of the retrospective cohort study on CUKA and RUKA. Abbreviations: CUKA, conventional unicompartmental knee arthroplasty; RUKA, robotic unicompartmental knee arthroplasty.

**Figure 2 jcm-11-00220-f002:**
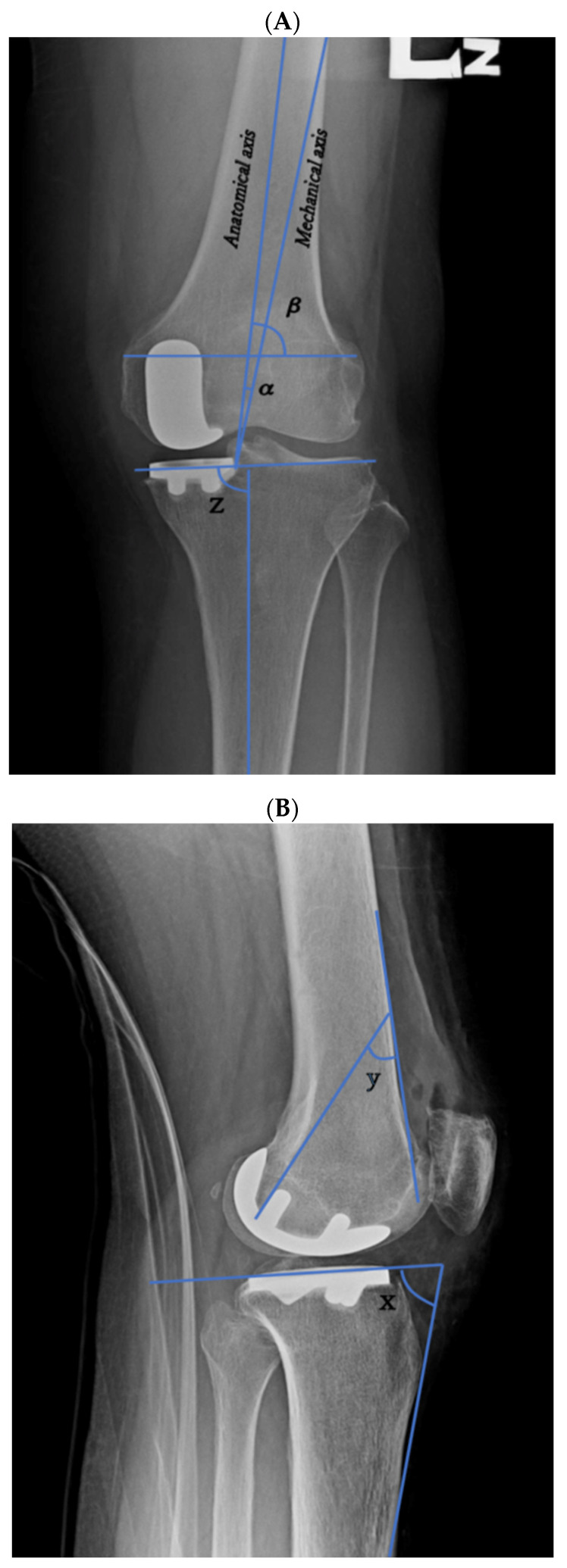
Radiograph parameter measurements of bone alignment and component positioning. (**A**) Anteroposterior view of the left knee indicating the measurement of anatomical axis angle α, femoral condyle angle β, and tibial axis angle z. (**B**) Lateral radiograph of the left knee indicating the measurement of femoral component angle *y* and tibial slope (90°—*x*).

**Figure 3 jcm-11-00220-f003:**
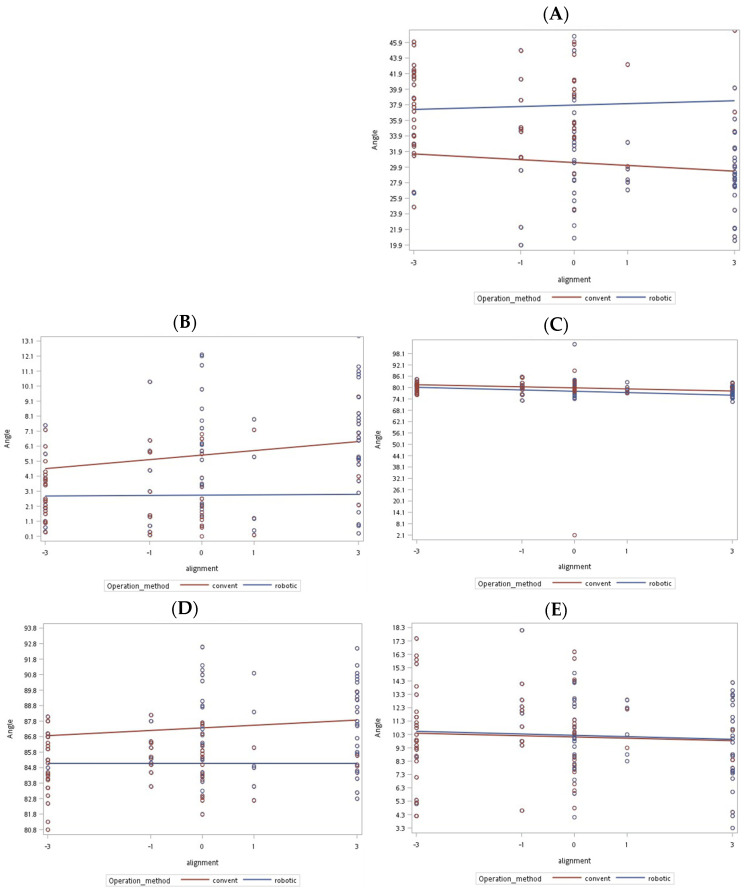
Distribution of the parameter measurements of postoperative bone alignment angles and implant positioning in both groups. (**A**) Femoral implant angle. (**B**) Postoperative anatomical axis angle. (**C**) Postoperative anatomical axis angle. (**D**) Postoperative femoral condyle angle. (**E**) Postoperative tibial slope angle. Abbreviations: convent = conventional.

**Table 1 jcm-11-00220-t001:** Preoperative and postoperative anatomical alignment parameters and implant positioning in the robotic-assisted and conventional group radiographs.

	Conventional (*n* = 61)	Robotic (*n* = 52)	*p* Value
Preanatomical axis angle (mean [SD])	3.26° ± 2.53°	3.57° ± 2.49°	0.519
Postanatomical axis angle (mean [SD])	5.87° ± 3.5°	2.81° ± 2.06°	<0.001
Prefemoral condyle angle (mean [SD])	81.21° ± 2.41°	81.12° ± 3.64°	0.88
Postfemoral condyle angle (mean [SD])	79.28° ± 4.19°	79.15° ± 11.22°	0.933
Femoral implant angle (mean [SD])	30.04° ± 5.65°	37.59° ± 5.4°	<0.001
Pretibial axis angle (mean [SD])	86.39° ± 2.93°	86.11° ± 2.56°	0.598
Posttibial axis angle (mean [SD])	87.57° ± 2.71°	85.07° ± 1.76°	<0.001
Pretibial slope angle (mean [SD])	12.48° ± 3.8°	11.35° ± 3.19°	0.093
Posttibial slope (mean [SD])	10° ± 3.19°	10.38° ± 3.32°	0.535
Tibial eminence saved (%)	22 (36.1%)	14 (26.9%)	0.403
Implant alignment aligned (%)	11 (18%)	30 (57.7%)	<0.001
Overhang or normal (%)			<0.001
Overhang	27 (44.3%)	2 (3.8%)	
Slight overhang	5 (8.2%)	2 (3.8%)	
Normal	21 (34.4%)	15 (28.8%)	
Normal, femoral implant abnormal	1 (1.6%)	0	
Normal, gap not clear	1 (1.6%)	0	
Slight underhang	3 (4.9%)	8 (15.4%)	
Underhang	3 (4.9%)	25 (48.1%)	

SD: standard deviation, *n*: number. *p* < 0.05 is statistically significant.

**Table 2 jcm-11-00220-t002:** Demographic characteristics and implant size in both groups.

	Conventional (*n* = 61)	Robotic (*n* = 52)	*p* Value
Age (years) (mean ± SD)	69.36 ± 9.14	68.52 ± 9.78	0.638
Sex (%)			0.521
Male	9 (14.8%)	11 (21.2%)	
Female	52 (85.2%)	41 (78.8%)	
ASA = 2 (%)	50 (81.97%)	48 (92.31%)	0.106
Operation site (%)			0.427
Left	30 (49.2%)	26 (50%)	
Right	31 (50.8%)	26 (50%)	
Medial or lateral = medial (%)	61 (100%)	52 (100%)	NA
Kellgren–Lawrence grade (mean [SD])	2.25 ± 0.43	2.77 ± 0.67	<0.001
Surgery Time (min) (mean [SD])	43.85 ± 6.08	60.87 ± 11.41	<0.001
Blood loss (ml)	201.2 ± 58.44	257.71 ± 100.85	<0.001
Pre-OP Hb (gm/dL) (mean [SD])	12.96 ± 1.05	12.86 ± 1.23	0.687
Post-OP Hb (gm/dL) (mean [SD])	11.23 ± 0.8	NM	NA
Complication	0	1 (1.9%)	0.936
Femoral implant size (%) (Femoral implant size: 1–6 indicates size of Stryker implant used in RUKA; A–E indicates size of Zimmer implant used in CUKA)			
1	0	9 (17.3%)	
2	0	13 (25%)	
3	0	19 (36.5%)	
4	0	7 (13.5%)	
5	0	2 (3.8%)	
6	0	2 (3.8%)	
A	1 (1.6%)	0	
B	10 (16.4%)	0	
C	32 (52.5%)	0	
D	15 (24.6%)	0	
E	3 (4.9%)	0	
Tibial implant size (mean [SD])	2.54 ± 0.74	2.94 ± 1.43	0.059

SD: standard deviation, *n*: number, min: minutes, mL: milliliter, gm/dL: grams per deciliter, OP: operation, Hb: hemoglobin. NA: not applicable, NM: not measured, ASA: American Society of Anesthesiologists physical status classification system. *p* < 0.05 is statistically significant.

**Table 3 jcm-11-00220-t003:** Follow-up preoperative and postoperative NRS, health, WOMAC, and EQ5D5L scores at 6 months, 1 year, and 2 years.

	CUKA (*n* = 38)	RUKA (*n* = 38)	*p* Value	CUKA (*n* = 38)	RUKA( *n* = 38)	*p* Value	CUKA (*n* = 38)	RUKA (*n* = 38)	*p* Value	CUKA (*n* = 38)	RUKA (*n* = 38)	*p* Value
	Pre-Op			6 m Post-OP			1 yr Post-OP			2 yr Post-OP		
Duration of pain (months) (mean [SD])	2.65 ± 1.04	2.66 ± 0.73	0.963									
NRS (mean [SD])	6.82 ± 1.31	6.79 ± 2.09	0.948	1.39 ± 1.79	1.76 ± 1.6	0.348	1.39 ± 1.79	1.82 ± 1.57	0.28	1.39 ± 1.79	1.82 ± 1.57	0.28
Health score (0–100) (mean [SD])	65.68 ± 7.56	71.68 ± 10.36	0.006	77.63 ± 7.51	79.03 ± 10.31	0.502	77.6 ± 7.51	78.7 ± 10.41	0.589	77.6 ± 7.51	78.7 ± 10.41	0.589
WOMAC total (mean [SD])	28.08 ± 8.13	27.64 ± 13.05	0.861	11.84 ± 12.36	14.44 ± 11.25	0.338	10.66 ± 11.66	14.64 ± 11.08	0.122	10.49 ± 11.44	14.67 ± 11.1	0.102
EQ5D5L total (mean [SD])	11.89 ± 13.79	8.85 ± 2.33	0.178	5.54 ± 2.09	5.9 ± 1.7	0.4	5.46 ± 2.1	6 ± 1.65	0.209	5.46 ± 2.1	6.03 ± 1.63	0.186

NRS: Numeric Rating Scale, WOMAC: Western Ontario and McMaster Universities Osteoarthritis Index, EQ5D5L: 5-level EQ-5D version. SD: standard deviation, *n*: number, OP: operation, m: month, yr: year. *p* < 0.05 is statistically significant.

## Data Availability

The data presented in this study are available on request from the corresponding author. The data are not publicly available because they contain the patients’ private information.

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
