# Peer review of "Comparison of Robotic and Conventional Unicompartmental Knee Arthroplasty Outcomes in Patients with Osteoarthritis: A Retrospective Cohort Study"

_jcm, 2021, doi:10.3390/jcm11010220_

Round 1

Reviewer 1 Report

Dear Authors,

as regard the M&M it will be necessary to describe who analysed the radiographic parameters and the observer relationship.

Furthermore why did not you use CT scan for radiographic measurements in consideration that you performed CT scan before and after knee prosthesis?

As regard discussion section i suggest to propose a solution to cope with the first limit of your study. In fact in order to improve the accuracy of your measurements and the femoral implant angle evaluation you could propose in a work in progress to use a CT scan citing the following article: Painful knee prosthesis: CT scan to assess patellar angle and implant malrotation

Spinarelli A . et al.

Muscles, Ligaments and Tendons Journal Open AccessVolume 6, Issue 4, Pages 461 - 4661 October 2016

Author Response

Response to reviewer 1 comments

Point 1:As regard the M&M it will be necessary to describe who analyzed the radiographic parameters and the observer relationship.

Response 1: The radiographic parameters were collected and analyzed by the first author. The first author, Christopher Wu, is a M.D graduate from Taipei Medical University and has been working for this study at our institute, Taipei Medical University Hospital.

Point 2: Furthermore why did not you use CT scan for radiographic measurements in consideration that you performed CT scan before and after knee prosthesis?

Response 2: Thank you for your input. Even though we used CT scans for radiographic measurements, some CT images were difficult to assess due to the interference of the metal artifacts. These metal artifacts could cover various bone–implant interface, surrounding tissue, resulting in inaccurate measurement. Thus, we decided to use X-ray since 2D level can avoid the interference of metal artifact. This allowed us to have a more consistent measurement instead of having to approximate on CT scans of certain locations that were blocked by these metal artifacts.

Point 3: As regard discussion section I suggest to propose a solution to cope with the first limit of your study. In fact in order to improve the accuracy of your measurements and the femoral implant angle evaluation you could propose in a work in progress to use a CT scan citing the following article: Painful knee prosthesis: CT scan to assess patellar angle and implant malrotation

Response 3: Thank you so much for your suggestion. We will cite the article which you provided to our discussion section for the purpose of improvement for the measurement evaluation. We proposed in a work in progress to use a CT scan as measurement assessment.

Reviewer 2 Report

Thank you for this interesting piece of scientific research. In Table 1, how do you explain and interpret the results regarding overhang and underhang between the two groups?

Language: please make sure a native English speaker re-reads and corrects the entire manuscript. All through the text there are errors as well as poor use of the English language, which can easily be optimized.

Author Response

Response to reviewer 2 comments

Point 1: In Table 1, how do you explain and interpret the results regarding overhang and underhang between the two groups?

Response1: Underhang and slightly underhang are considered as acceptable component positioning. Whereas overhang and slightly overhang are considered as inaccurate component positioning. As a result, we observed that there were more inaccurately positioned components (overhang and slightly overhang) in CUKA compared to RUKA. (44.3% + 8.2 % in CUKA group vs 3.8% + 3.8% in RUKA group, p<0.001). In CUKA, more overhang could result in an increased risk of knee pain due to soft tissue impingement on edges. On the other hand, underhang components do not cause soft tissue issues therefore it can be interpreted as better outcome.

Point 2: Language: please make sure a native English speaker re-reads and corrects the entire manuscript. All through the text there are errors as well as poor use of the English language, which can easily be optimized.

Response 2: Thank you for your recommendation. We will ask the editor who is a native English speaker to correct our entire manuscript. We apologize for the errors as well as poor use of the English language in the manuscript.